# Global Convergence of Multi-Agent Policy Gradient in Markov Potential Games

## Abstract

Potential games are arguably one of the most important and widely studied classes of normal form games. They define the archetypal setting of multi-agent coordination as all agent utilities are perfectly aligned with each other via a common potential function. Can this intuitive framework be transplanted in the setting of Markov Games? What are the similarities and differences between multi-agent coordination with and without state dependence? We present a novel definition of Markov Potential Games (MPG) that generalizes prior attempts at capturing complex stateful multi-agent coordination. Counter-intuitively, insights from normal-form potential games do not carry over as MPGs can consist of settings where state-games can be zero-sum games. In the opposite direction, Markov games where every state-game is a potential game are not necessarily MPGs. Nevertheless, MPGs showcase standard desirable properties such as the existence of deterministic Nash policies. In our main technical result, we prove fast convergence of independent policy gradient to Nash policies by adapting recent gradient dominance property arguments developed for single agent MDPs to multi-agent learning settings.

## 1   Introduction

Reinforcement learning (RL) has been a fundamental driver of numerous recent advances in Artificial Intelligence (AI) applications that range from super-human performance in competitive game-playing [28, 29, 5] and strategic decision-making in multiple tasks [21, 23, 33] to robotics, autonomous-driving and cyber-physical systems [6, 37]. A core ingredient for the success of single-agent RL systems, which are typically modelled as Markov Decision Processes (MDPs), is the guarantee of existence of stationary deterministic optimal policies [3, 30]. This allows for the design of efficient algorithms that provably converge towards the optimal policy [1]. However, a majority of the above systems involve multi-agent interactions and despite the notable empirical advancements, there is a lack of understanding about the theoretical convergence guarantees of the existing multi-agent reinforcement learning (MARL) algorithms.

The main challenge in the transition from single to multi-agent RL settings is the computation of *Nash policies*. A Nash policy for $n > 1$ agents is defined to be a profile of policies $(\pi_1^*, ..., \pi_n^*)$ so that by fixing the stationary policies of all agents but $i$, $\pi_i^*$ is an optimal policy for the resulting single-agent MDP and this is true for all $1 \leq i \leq n$ [1] (see Definition 1). Note that in multi-agent settings, Nash policies *may not be unique* in principle.

A common approach for computing Nash policies in MDPs is the use of *policy gradient* methods. The significant progress in the analysis of such methods during the last couple of years, including [1] (and references therein), mainly concerns the single-agent case: the convergence properties of

---

[1]Analogue of Nash equilibrium notion.

policy gradient in MARL remain poorly understood. Existing steps towards a theory for multi-agent settings involve the papers of [10] who show convergence of *independent policy gradient* to the optimal policy, for two-agent zero-sum stochastic games, of [36] who improve the result of [10] using optimistic policy gradient and of [38] who study extensions of Natural Policy Gradient using function approximation. It is worth noting that the positive results of [10, 36] and [38] depend on the fact that two-agent stochastic zero-sum games satisfy the "min-max equals max-min" property [27] (even though the value-function landscape may not be convex-concave, which implies that Von Neumann's celebrated minimax theorem may not be applicable).

**Model and Informal Statement of Results.** While the previous works make progress in *competitive* interactions, i.e., interactions in which gains can only come at the expense of others, MARL in *cooperative* settings remains largely under-explored and constitutes one of the current frontiers in AI research [9, 8]. Based on this, our work is motivated by the following natural question:

> *Can we get (provably) fast convergence guarantees for multi-agent RL settings*
> *in which cooperation is desirable?*

To address this question, we define and study a class of $n$-agent MDPs that naturally generalize normal form potential games [22], called *Markov Potential Games (MPGs)*. In words, a multi-agent MDP is a MPG as long as there exists a (state-dependent) real-valued potential function $\Phi$ so that if an agent $i$ changes their policy (and the rest of the agents keep their policy unchanged), the difference in agent $i$'s value/utility, $V^i$, is captured by the difference in the value of $\Phi$ (see Definition 2). Weighted and ordinal MPGs are defined similar to the normal form counterparts (see Remark 1).

Under our definition, we answer the above motivating question in the affirmative. In particular, we show that if every agent $i$ independently runs (with simultaneous updates) policy gradient on his utility/value $V^i$, after $O(1/\epsilon^2)$ iterations, the system will reach an $\epsilon$-approximate Nash policy (see informal Theorem 1.1 and formal Theorem 4.2). Moreover, we show the finite sample analogue, that is if every agent $i$ independently runs (with simultaneous updates) stochastic policy gradient, then with high probability, the system will reach an $\epsilon$-approximate Nash policy after $O(1/\epsilon^6)$ iterations.

Along the way, we prove several properties about the structure of MPGs and their Nash policies (see Theorem 1.2 and Section 3). Our results can be summarized in the following two Theorems.

**Theorem 1.1** (Convergence of Policy Gradient (Informal))**.** *Consider a MPG with $n$ agents and let $\epsilon > 0$. (a) If each agent $i$ runs independent policy gradient using direct parameterization on his policy and that the updates are simultaneous, then, the learning dynamics reach an $\epsilon$-Nash policy after $\mathcal{O}(1/\epsilon^2)$ iterations. (b) If each agent $i$ runs stochastic policy gradient using greedy parameterization (see (3)) on his policy and the updates are simultaneous, then the learning dynamics reach an $\epsilon$-Nash policy after $\mathcal{O}(1/\epsilon^6)$ iterations.*

This result holds trivially for weighted MPGs and asymptotically also for ordinal MPGs, see Remark 2.

**Theorem 1.2** (Structural Properties of MPGs)**.** *The following facts are true for MPGs with $n$-agents:*

*(a) There always exists a Nash policy profile $(\pi_1^*, \ldots, \pi_n^*)$ so that $\pi_i^*$ is deterministic for each agent $i$ (see Theorem 3.1).*
*(b) We can construct MDPs for which each state is an underlying potential game but the MDPs are not MPGs. This can be true regardless of whether the whole MDP is competitive or cooperative in nature (see Examples 1 and 2, respectively). On the opposite side, we can construct MDPs that are MPGs but which include states that are purely competitive (i.e., zero-sum games), see Example 3.*
*(c) We provide sufficient conditions so that a MDP is a MPG. These include cases where each state is an underlying potential game and the transition probabilities are not affected by agents actions or the reward functions satisfy certain regularity conditions between different states (see conditions C1 and C2 in Proposition 3.2).*

**Technical Overview.** The first challenge in the proof of Theorem 1.1 is that multi-agent settings (MPGs) do not satisfy the gradient dominance property, which is an important part in the proof of convergence of policy gradient in single-agent settings [1]. In particular, there is no uniqueness of optimal policies and as a result, there is not a properly defined notion of value in MPGs (in contrast to zero-sum stochastic games [10]). On the positive side, we show that agent-wise (i.e., after fixing the policy of all agents but $i$), the value function, $V^i$, satisfies the gradient dominance property along the direction of $\pi_i$ (policy of agent $i$). This can be leveraged to show that every *(approximate) stationary*

*point* (Definition 4) of the potential function $\Phi$ is an *(approximate) Nash policy* (Lemma 4.1). As a result, convergence to an approximate Nash policy is established by showing that $\Phi$ is smooth and then applying *Projected Gradient Ascent* (PGA) on $\Phi$. This step uses the rather well-known fact that (PGA) converges to $\epsilon$-stationary points in $O(1/\epsilon^2)$ iterations for smooth functions. As a result, by applying PGA on the potential $\Phi$, one gets an approximate Nash policy. Our convergence result then follows by showing that PGA on the potential function, $\Phi$, generates the same dynamics as if each agent $i$ runs independent PGA on their value function, $V^i$.

In the case that agents do not have access to exact gradients, we derive a similar result for finite samples. In this case, we apply *Projected Stochastic Gradient Ascent (PSGA)* on $\Phi$ which (as was the case for PGA) can be shown to be the same as when agents apply PSGA independently on their individual value functions. The key is to get an *unbiased sample* for the gradient of the value functions and prove that it has bounded variance (in terms of the parameters of the MPG). This comes from the discount factor, $\gamma$; in this case, $1 - \gamma$ can be interpreted as the probability to terminate the MDP at a particular state (and $\gamma$ to continue). This can be used to show that a trajectory of the MDP is an unbiased sample for the gradient of the value functions. To guarantee that the estimate has bounded variance, we apply the approach of [10] which requires that agents perform PSGA with $\alpha$-greedy exploration (see (3)). The main idea is that this parameterization stays away from the boundary of the simplex throughout its trajectory.

Concerning our structural results in Theorem 1.2, the main challenge is (again) the lack of a value in general multi-agent settings and the dependence of state-transitions (in addition to agents' rewards) on agents' actions. The proof of Theorem 3.1 shows that these issues can be still successfully handled within the class of MPGs by studying single-agent deviations (to deterministic optimal policies) which keep the value of the potential constant (at its global maximum). Our examples in this part show that the class of MPGs can be significantly larger than state based potential games but also that even simple coordination games may fail to satisfy the (exact) MPG property.

## 2  Preliminaries

**Markov Decision Process (MDP).**   The following notation is standard and largely follows [1] and [10]. We consider a setting with $n$ agents who repeatedly select actions in a shared Markov Decision Process (MDP). The goal of each agent is to maximize their respective value function. Formally, a MDP is defined as a tuple $\mathcal{G} = (\mathcal{S}, \mathcal{N}, \{\mathcal{A}_i, R_i\}_{i \in \mathcal{N}}, P, \gamma, \rho)$, where $\mathcal{S}$ is a finite state space of size $S = |\mathcal{S}|$, $\mathcal{N} = \{1, 2, \ldots, n\}$ is a the set of active agents in the MDP and $\mathcal{A}_i$ is a finite action space of size $A_i = |\mathcal{A}_i|$ for each agent $i \in \mathcal{N}$ with generic element $a_i \in \mathcal{A}_i$. We will write $\mathcal{A} = \prod_{i \in \mathcal{N}} \mathcal{A}_i$ and $\mathcal{A}_{-i} = \prod_{j \neq i} \mathcal{A}_j$ to denote the joint action spaces of all agents and of all agents other than $i$ with generic elements $\mathbf{a} = (a_i)_{i \in \mathcal{N}}$ and $\mathbf{a_{-i}} = (a_j)_{j \neq i}$, respectively. $R_i : \mathcal{S} \times \mathcal{A} \to [-1, 1]$ is the individual reward function of agent $i \in \mathcal{N}$, i.e., $R_i(s, a_i, \mathbf{a}_{-i})$ is the instantaneous reward of agent $i$ when agent $i$ takes action $a_i$ and all other agents take actions $\mathbf{a}_{-i}$ at state $s \in \mathcal{S}$. $P$ is the transition probability function, for which $P(s' \mid s, \mathbf{a})$ is the probability of transitioning from $s$ to $s'$ when $\mathbf{a} \in \mathcal{A}$ is the action profile chosen by the agents. Finally, $\gamma$ is a discount factor for future rewards of the MDP, shared by all agents and $\rho \in \Delta(\mathcal{S})$ is a distribution for the initial state at time $t = 0$.[2]

Whenever time is relevant, we will index the above terms with $t$. In particular, at each time step $t \geq 0$, all agents observe the state $s_t \in \mathcal{S}$, select actions $\mathbf{a}_t = (a_{i,t}, \mathbf{a}_{-i,t})$, receive rewards $r_{i,t} := R_i(s_t, \mathbf{a}_t), i \in \mathcal{N}$ and transition to the next state $s_{t+1} \sim P(\cdot \mid s_t, \mathbf{a}_t)$. We will write $\tau = (s_t, \mathbf{a}_t, \mathbf{r}_t)_{t \geq 0}$ to denote the trajectories of the system, where $\mathbf{r}_t := (r_{i,t}), i \in \mathcal{N}$.

**Policies and Value Functions.**   For each agent $i \in \mathcal{N}$, a deterministic, stationary policy $\pi_i : \mathcal{S} \to \mathcal{A}_i$ specifies the action of agent $i$ at each state $s \in \mathcal{S}$, i.e., $\pi_i(s) = a_i \in \mathcal{A}_i$ for each $s \in \mathcal{S}$. A stochastic, stationary policy $\pi_i : \mathcal{S} \to \Pi_i$, where $\Pi_i := \Delta(\mathcal{A}_i)^S$, specifies a probability distribution over the actions of agent $i$ for each state $s \in \mathcal{S}$. In this case, we will write $a_i \sim \pi_i(\cdot \mid s)$ to denote the randomized action of agent $i$ at state $s \in \mathcal{S}$. As above, we will write $\pi = (\pi_i)_{i \in \mathcal{N}} \in \Pi := \times_{i \in \mathcal{N}} \Delta(\mathcal{A}_i)^S$ and $\pi_{-i} = (\pi_j)_{i \neq j \in \mathcal{N}} \in \Pi_{-i} := \times_{i \neq j \in \mathcal{N}} \Delta(\mathcal{A}_j)^S$ to denote the joint policies of all agents and of all agents other than $i$, respectively. A joint policy $\pi$ induces a distribution $\text{Pr}^\pi$ over trajectories $\tau = (s_t, \mathbf{a}_t, \mathbf{r}_t)_{t \geq 0}$, where $s_0$ is drawn from the initial state distribution $\rho$ and $a_{i,t}$ is drawn from $\pi_i(\cdot \mid s_t)$ for all $i \in \mathcal{N}$.

---

[2]We will write $\Delta(\mathcal{X})$ to denote the set of probability distributions over any set $\mathcal{X}$.

The value function, $V_s^i : \Pi \to \mathbb{R}$, gives the expected reward of agent $i \in \mathcal{N}$ when $s_0 = s$ and the agents draw their actions, $\mathbf{a}_t = (a_{i,t}, \mathbf{a}_{-i,t})$, at time $t \geq 0$ from policies $\pi = (\pi_i, \pi_{-i})$

$$V_s^i(\pi) := \mathbb{E}_\pi \left[ \sum_{t=0}^\infty \gamma^t r_{i,t} \mid s_0 = s \right]. \tag{1}$$

We also denote $V_\rho^i(\pi) = \mathbb{E}_{s \sim \rho} \left[ V_s^i(\pi) \right]$ if the initial state is random and follows distribution $\rho$. The solution concept that we will be focusing on are the Nash Policies. Formally:

**Definition 1** ($\epsilon$-Nash Policy). A joint policy $\pi^* = (\pi_i^*)_{i \in \mathcal{N}}$ is an $\epsilon$-Nash policy if there exists an $\epsilon \geq 0$ so that for each agent $i \in \mathcal{N}$, $V_s^i(\pi_i^*, \pi_{-i}^*) \geq V_s^i(\pi_i, \pi_{-i}^*) - \epsilon$, for all $\pi_i \in \Delta(\mathcal{A}_i)^S$, and all $s \in \mathcal{S}$. If $\epsilon = 0$, then $\pi^*$ is a called a *Nash policy*. In this case, $\pi_i^*$ maximizes each agent $i$'s value function for each starting state $s \in \mathcal{S}$ given the policies, $\pi_{-i}^* = (\pi_j^*)_{j \neq i}$, of all other agents $j \neq i \in \mathcal{N}$. The definition of a Nash policy remains the same if $s \sim \rho$ (random starting state).

# 3 Markov Potential Games

We are now ready to define the class of MDPs that we will focus on for the rest of the paper, i.e., Markov Potential Games.

**Definition 2** (Markov Potential Game). A Markov Decision Process (MDP), $\mathcal{G}$, is called a *Markov Potential Game (MPG)* if there exists a (state-dependent) function $\Phi_s : \Pi \to \mathbb{R}$ for $s \in \mathcal{S}$ so that

$$\Phi_s(\pi_i, \pi_{-i}) - \Phi_s(\pi_i', \pi_{-i}) = V_s^i(\pi_i, \pi_{-i}) - V_s^i(\pi_i', \pi_{-i}),$$

for all agents $i \in \mathcal{N}$, all states $s \in \mathcal{S}$ and all policies $\pi_i, \pi_i' \in \Pi_i, \pi_{-i} \in \Pi_{-i}$. We should note that by linearity of expectation, it follows that $\Phi_\rho(\pi_i, \pi_{-i}) - \Phi_\rho(\pi_i', \pi_{-i}) = V_\rho^i(\pi_i, \pi_{-i}) - V_\rho^i(\pi_i', \pi_{-i})$, where $\Phi_\rho(\pi) := \mathbb{E}_{s \sim \rho} [\Phi_s(\pi)]$.

As in normal-form games, an immediate consequence of this definition is that the value function of each agent in a MPG can be written as a sum of the potential *(common term)* and a term that does not depend on that agent's policy *(dummy term)*, cf. Proposition B.1 in Appendix B, i.e., for each agent $i \in \mathcal{N}$ there exists a function $U_s^i : \Pi_{-i} \to \mathbb{R}$ so that $V_s^i(\pi) = \Phi_s(\pi) + U_s^i(\pi_{-i})$, for all $\pi \in \Pi$.

*Remark* 1 (Ordinal and Weighted Potential Games). Similar to normal-form games, we may also define more general notions of MPGs, such as *weighted* or *ordinal* MPGs. Specifically, if there exist positive constants $w_i > 0, i \in \mathcal{N}$ so that

$$\Phi_s(\pi_i, \pi_{-i}) - \Phi_s(\pi_i', \pi_{-i}) = w_i(V_s^i(\pi_i, \pi_{-i}) - V_s^i(\pi_i', \pi_{-i})),$$

then $\mathcal{G}$ is called a *Weighted Markov Potential Game (WMPG)*. If for all agents $i \in \mathcal{N}$, all states $s \in \mathcal{S}$ and all policies $\pi_i, \pi_i' \in \Pi_i, \pi_{-i} \in \Pi_{-i}$, the function $\Phi_s, s \in \mathcal{S}$ satisfies

$$\Phi_s(\pi_i, \pi_{-i}) - \Phi_s(\pi_i', \pi_{-i}) > 0 \iff V_s^i(\pi_i, \pi_{-i}) - V_s^i(\pi_i', \pi_{-i}) > 0,$$

then the MPD, $\mathcal{G}$, is called an *Ordinal Markov Potential Game (OMPG)*.

Similarly to normal-form games, such classes are naturally motivated also in the setting of multi-agent MDPs. As Example 2 shows, even simple potential-like settings, i.e., settings in which coordination is desirable for all agents, may fail to be exact MPGs (but may still be ordinal or weighted MPGs). From our current perspective, ordinal and weighted MPGs remain relevant, since our main convergence results on the convergence of policy gradient carry over (in an exact or asymptotic sense) also in these classes of games (see Remark 2). As with the rest of the proofs (and technical details) of Section 3, the proof of Theorem 3.1 is provided in Appendix B.

**Existence of Deterministic Nash Policies in MPGs.** Before studying which types of MDPs are captured by Definition 2, we first show that MPGs always possess deterministic Nash policies (similarly to their single-state counterparts, i.e., normal-form potential games [22]). This is established in Theorem 3.1, which settles part (a) of Theorem 1.2

**Theorem 3.1** (Deterministic Optimal Policy Profile). *Let $\mathcal{G}$ be a Markov Potential Game (MPG). Then, there exists a Nash policy $\pi^* \in \Delta(\mathcal{A})^S$ which is deterministic, i.e., for each agent $i \in \mathcal{N}$ and each state $s \in \mathcal{S}$, there exists an action $a_i \in \mathcal{A}_i$ so that $\pi_i^*(a_i \mid s) = 1$.*

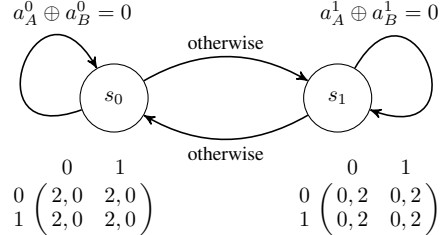

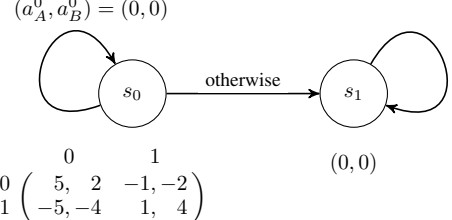

$$a_A^0 \oplus a_B^0 = 0 \qquad a_A^1 \oplus a_B^1 = 0$$

$$
\begin{array}{c}
\begin{array}{cc} 0 & 1 \end{array} \\
\begin{array}{c} 0 \\ 1 \end{array}
\begin{pmatrix} 2,0 & 2,0 \\ 2,0 & 2,0 \end{pmatrix}
\end{array}
\qquad
\begin{array}{c}
\begin{array}{cc} 0 & 1 \end{array} \\
\begin{array}{c} 0 \\ 1 \end{array}
\begin{pmatrix} 0,2 & 0,2 \\ 0,2 & 0,2 \end{pmatrix}
\end{array}
$$

$$(a_A^0, a_B^0) = (0,0)$$

$$
\begin{array}{c}
\begin{array}{cc} 0 & 1 \end{array} \\
\begin{array}{c} 0 \\ 1 \end{array}
\begin{pmatrix} 5,\ 2 & -1,-2 \\ -5,-4 & 1,\ 4 \end{pmatrix}
\end{array}
\qquad
(0,0)
$$

Figure 1: A MDP with normal-from potential games at each state (shown in matrix form below each state) but which is not a MPG due to conflicting preferences over states.

Figure 2: A MDP with normal-form potential games at each state which is an ordinal MPG but not a MPG despite common preferences over states.

Starting from an arbitrary Nash policy profile that is also a global maximizer of the potential function, the proof of Theorem 3.1 (which is deferred to Appendix B) relies on an iterative reduction process of its non-deterministic components. At each iteration, we isolate an agent $i \in \mathcal{N}$, and find a deterministic (optimal) policy for that agent in the (single-agent) MDP in which the policies of all other agents but $i$ remain fixed. The important observation is that the resulting profile is again a global maximizer of the potential and hence, a Nash policy profile. This argument critically relies on the MPG structure and does not seem directly generalizable to MDPs that do not satisfy Definition 2.

**Sufficient Conditions for MPGs.** Based on the above, it is tempting to think that MDPs which are potential at every state (meaning that the immediate rewards at every state are captured by a (normal-form) potential game at that state) are trivially MPGs. As we show in Examples 1 and 2, this intuition fails in the most straightforward way: we can construct simple MDPs that are potential at every state but which are purely competitive (do not possess a deterministic Nash policy) overall (Example 1) or which are cooperative in nature overall but which do not possess an exact potential function (Example 2).

**Example 1.** Consider the MDP in Figure 1. To show that $\mathcal{G}$ is not a MPG, it suffices to show that it cannot have a deterministic optimal policy as should be the case according to Theorem 3.1. To obtain a contradiction, assume that agent $A$ is using a deterministic action $a_A^0 \in \{0, 1\}$ at state 0. Then, agent $B$, who prefers to move to state 1, will optimize their utility by choosing the action $a_B^0 \in \{0, 1\}$ that yields $a_A^0 \oplus a_B^0 = 1$. In other words, given any deterministic action of agent $A$ at state 0, agent $B$ can choose an action that always moves the sequence of play to state 1. Thus, such an action cannot be optimal for agent $A$ which implies that the MDP $\mathcal{G}$ does not have a deterministic optimal policy profile as claimed.

Intuitively, competition arises in Example 1 because the two agents play a game of *matching pennies* in terms of the states that they prefer (which can be determined by the actions that they choose) despite the fact that the immediate rewards at each state are determined by normal form potential games. Example 2 shows that a state-based potential game may fail to be a MPG even if agents have similar preferences over states.

**Example 2.** In $s_0$ the agents play a Battle of the Sexes game and hence a potential game, while in $s_1$ they receive no reward (which is trivially a potential game). A simple calculation shows that there is not an exact potential function due to the dependence of the transitions on agents' actions (thus, this MDP is not a MPG). However, in the case of Example 2, it is straightforward to show that the game is an ordinal potential game, cf. Appendix B.1.

The previous discussion focuses on games that consist of normal-form potential games at every state, which leaves an important question unanswered: are there games which are not potential at every state but which are captured by the current definition of MPGs? Example 3 (see Figure 3) answers this question affirmatively. Together with Example 1, this settles the claim in Theorem 1.2, part (b).

**Proposition 3.2** (Sufficient Conditions for MPGs). *Consider a MDP $\mathcal{G}$ in which every state $s \in \mathcal{S}$ is a potential game, i.e., the immediate rewards $R(s, \mathbf{a}) = (R_i(s, \mathbf{a}))_{i \in \mathcal{N}}$ for each state $s \in \mathcal{S}$ are captured by the utilities of a potential game with potential function $\phi_s$. Additionally, assume that one of the following conditions holds*

*C1. Agent-Independent Transitions: $P(s' \mid s, \mathbf{a})$ does not depend on $\mathbf{a}$, that is, $P(s' \mid s, \mathbf{a}) = P(s' \mid s)$ is just a function of the present state for all states $s, s' \in \mathcal{S}$.*

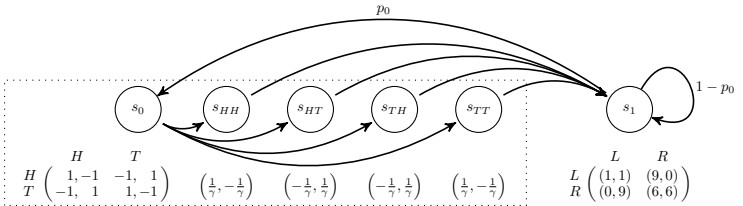

Figure 3: A 2-player MDP which is not potential at every state but which is overall an MPG. While state $s_1$ corresponds to a zero-sum game, the states inside the dotted rectangle do form a potential game which can be used to show the MPG property whenever $p_0$ does not depend on agents' actions.

*C2. Equality of Individual Dummy Terms: $P(s' \mid s, \mathbf{a})$ is arbitrary but the dummy terms of each agent's immediate rewards are equal across all states, i.e., there exists a function $u^i : \Delta(\mathcal{A}_{-i})^S \to \mathbb{R}$ such that $R_i(s, a_i, \mathbf{a}_{-i}) = \phi_s(\pi_i, \pi_{-i}) + u^i(\pi_{-i})$, for all states $s \in \mathcal{S}$.*

*If either C1 or C2 are true, then $\mathcal{G}$ is a MPG.*

**Relation to Other Works on MPGs**  Condition C2 (or variations of it) is also known as *state-transitivity* and is present as requirement in the existing definitions of potential-like MDPs, see e.g., [16, 19, 20] and along with some additional conditions on the transitions also in [32]. Example 3 shows that such conditions are restrictive, in the sense that they do not capture simple MDPs that intuitively have a cooperative structure. Similarly, Example 2 motivates the study of weighted or ordinal MPGs (cf. Remark 1). As we show, our convergence results about independent policy gradient naturally apply to these classes as well (see Remark 2).

Another sufficient condition for a MPD that is potential at every state to be a MPG is that the instantaneous rewards of all agents are the same at each state, i.e., that $R_i(s, a_i, \mathbf{a}_{-i}) = \phi_s(a_i, \mathbf{a}_{-i})$ for all agents $i \in \mathcal{N}$, all actions $a_i \in \mathcal{A}_i$ and all states $s \in \mathcal{S}$. MDPs that satisfy this condition are called *Team Markov Games* and their analysis trivially boils down to single agent settings. However, they constitute the only (to the best of our knowledge) cooperative multi-agent setting (covered by MPGs) that have been successfully addressed in terms of convergence of independent policy gradient prior to this work, [35].

## 4   Convergence of Policy Gradient in Markov Potential Games

The current section presents the main lemmas and steps for the proof of convergence of (projected) policy gradient (and its stochastic variant) to approximate Nash policies in Markov Potential Games (MPGs). We analyze these cases using direct and $\alpha$-greedy parameterization, respectively. All proofs and auxiliary materials are deferred to the supplementary material (full version).

**Independent Policy Gradient and Direct Parameterization.**  We assume that all agents update their policies *independently* according to the *projected gradient ascent (PGA)* or *policy gradient* algorithm. Independence here refers to the fact that (PGA) requires only local information (each agent's own rewards, actions and view of the environment) to determine the updates. Such protocols are naturally motivated in distributed AI settings in which all information about the interacting agents, the type of interaction and the agent's actions (policies) is encoded in the environment of each agent.[3] The PGA algorithm is given by

$$\pi_i^{(t+1)} := P_{\Delta(\mathcal{A}_i)^S}\left(\pi_i^{(t)} + \eta \nabla_{\pi_i} V_\rho^i(\pi^{(t)})\right), \tag{PGA}$$

for each agent $i \in \mathcal{N}$, where $P_\Delta(\mathcal{A}_i)^S$ is the projection onto $\Delta(\mathcal{A}_i)^S$ in the Euclidean norm. Here, the additional argument $t \geq 0$ denotes time. We also assume that all players $i \in \mathcal{N}$ use direct policy parameterizations, i.e., $\pi_i(a \mid s) = x_{i,s,a}$, with $x_{i,s,a} \geq 0$ for all $s \in \mathcal{S}, a \in \mathcal{A}_i$ and $\sum_{a \in \mathcal{A}_i} x_{i,s,a} = 1$ for all $s \in \mathcal{S}$. This parameterization is complete in the sense that any stochastic policy can be represented in this class [1].

---

[3]In practice, even though each agent treats their environment as fixed, the environment changes as other agents update their policies. This makes the analysis of such protocols particularly challenging in general and highlights the importance of studying classes of MDPs in which convergence can be obtained.

In practice, agents use *projected stochastic gradient ascent* (PSGA), according to which, the actual gradient, $\nabla_{\pi_i} V_\rho^i(\pi^{(t)})$, is replaced by an estimate thereof that is calculated from a randomly selected (yet finite) sample of trajectories of the MDP. This estimate, $\hat{\nabla}_{\pi_i}^{(t)}$ may be derived from a single or a batch of observations which in expectation behave as the actual gradient. We choose the estimate of the gradient of $V_\rho^i$ to be

$$\hat{\nabla}_{\pi_i}^{(t)} = R_i^{(T,t)} \sum_{k=0}^{T} \nabla \log \pi_i(a_k^{(t)} \mid s_k^{(t)}), \tag{2}$$

where $s_0^t \sim \rho$, and $R_i^{(T,t)} = \sum_{k=0}^{T} r_{i,t}^k$ is the sum of rewards of agent $i$ for a batch of time horizon $T$ along the trajectory generated by the stochastic gradient ascent algorithm at its $t$-th iterate.

The direct parameterization is not sufficient to ensure that the variance of the gradient estimator is bounded (as policies approach the boundary). In this case, we will require that each agent $i \in \mathcal{N}$ uses instead direct parameterization with $\alpha$-greedy exploration as follows

$$\pi_i(a \mid s) = (1 - \alpha_i)x_{i,s,a} + \alpha/A_i, \tag{3}$$

where $\alpha$ is the exploration parameter for all agents. Under greedy exploration, it can be shown that (2) is unbiased and has bounded variance for $\alpha$-greedy exploration (see Lemma 4.3). The form of PSGA is given below:

$$\pi_i^{(t+1)} := P_{\Delta(\mathcal{A}_i)^S} \left( \pi_i^{(t)} + \eta \hat{\nabla}_{\pi_i}^{(t)} \right). \tag{PSGA}$$

**Proofs of main results.** The first step is to observe that, in MPGs, the (partial) derivatives of the value functions and the potential function are equal, i.e., $\nabla_{\pi_i} V_s^i(\pi) = \nabla_{\pi_i} \Phi(\pi)$ for all $i \in \mathcal{N}$ (property P2 in Proposition B.1). Together with the separability of the projection operator, i.e., the fact that projecting independently for each agent $i$ on $\Delta(\mathcal{A}_i)^S$ is the same as jointly projecting on $\Delta(\mathcal{A})^S$ (see Lemma 4.1), this establishes that running (PGA) or (PSGA) on each agent's value function is equivalent to running (PGA) or (PSGA) on the potential function $\Phi$.

Based on the above, the next step is to study the stationary points of $\Phi$. Lemma 4.1 suggests that as long as policy gradient reaches a point $\pi^{(t)}$ with small gradient along the directions in $\Delta(\mathcal{A})^S$, it must be the case that $\pi^{(t)}$ is an approximate Nash policy.

**Lemma 4.1** (Stationarity of $\Phi$ implies Nash). *Let $\epsilon \geq 0$, $\pi$ be an $\epsilon$-stationary point of $\Phi$ (see Definition 4). Then, it holds that $\pi$ is a $\frac{\sqrt{S}D\epsilon}{1-\gamma}$-Nash policy.*

Lemma 4.1 will be the one of two mains ingredients to establish convergence of (PGA) and (PSGA). To prove Lemma 4.1, we will use an agent-wise version of the "Gradient Domination property", that has been shown to hold in single-agent MDPs [1] (see Lemma 4.3). The second main ingredient is the fact that $\Phi$ is a $\beta$-smooth function (its gradient is Lipschitz) with parameter $\beta = \frac{2n\gamma A_{\max}}{(1-\gamma)^3}$.

**Exact gradients case.** Theorem 1.1 (restated formally below) about rates of convergence of (PGA) can now be proved following standard arguments (in particular an ascent property, Lemma D.1), on analysis of convergence of gradient descent to approximate stationary points in non-convex optimization [11]. The *ascent lemma* suggests that for any $\beta$-smooth function, $f$, it holds that $f(x') - f(x) \geq \frac{1}{2\beta} \|x' - x\|_2^2$, where $x'$ is the next iterate of (PGA). Thus, having shown that $\Phi$ is a $\beta$-smooth function, the ascent lemma implies in our setting that

$$\Phi_\mu(\pi^{(t+1)}) - \Phi_\mu(\pi^{(t)}) \geq \frac{(1-\gamma)^3}{4\gamma A_{\max} n} \left\| \pi^{(t+1)} - \pi^{(t)} \right\|_2^2. \tag{4}$$

Putting everything together, we can show the following theorem.

**Theorem 4.2** (Formal Theorem 1.1, part (a)). *Let $\mathcal{G}$ be a MPG and let $s_0 \in \mathcal{S}$ denote an arbitrary initial state. Let also $A_{\max} = \max_i |\mathcal{A}_i|$, and set the number of iterations to be $T = \frac{16\gamma n D^2 S A_{\max}}{(1-\gamma)^5 \epsilon^2}$ and the learning rate (step-size) to be $\eta = \frac{(1-\gamma)^3}{2\gamma A_{\max} n}$. If the agents run independent projected policy gradient (PGA) starting from arbitrarily initialized policies, then there exists a $t \in \{1, \ldots, T\}$ such that $\pi^{(t)}$ is an $\epsilon$-approximate Nash policy.*

**Finite samples case.** In the case of finite samples, we analyze (PSGA) on the value $V^i$ of each agent $i$ which (as was the case for PGA) can be shown to be the same as applying projected gradient ascent on $\Phi$. In this case, we choose $\alpha$-greedy parametrization with $\alpha$ chosen appropriately. The key is to get an estimate of the gradient of $\Phi$ (see (2)) at every iterate. Lemma 4.3 argues that the estimator of equation (2) is unbiased and has bounded variance.

**Lemma 4.3** (Unbiased estimator with bounded variance ). *It holds that $\hat{\nabla}_{\pi_i}^{(t)}$ is an unbiased estimator of $\nabla_{\pi_i}\Phi$ with bounded variance for all $i \in \mathcal{N}$, i.e.,*

$$\mathbb{E}_{\pi^{(t)}}\hat{\nabla}_{\pi_i}^{(t)} = \nabla_{\pi_i}\Phi_\mu(\pi^{(t)}), \ \ with \ \ \mathbb{E}_{\pi^{(t)}}\left\|\hat{\nabla}_{\pi_i}^{(t)}\right\|_2^2 \leq \frac{24A_{\max}^2}{\epsilon(1-\gamma)^4}, for \ all \ i \in \mathcal{N}.$$

In this case, $1 - \gamma$ captures the probability for the MDP to terminate after each round since we consider finite length trajectories. Using the above, we can now state part (b) of Theorem 1.1. Together with Lemma 4.3 and the stationarity-Lemma (Lemma 4.1), i.e., that stationary points of $\Phi$ are Nash policies, its proof uses the smoothness of $\Phi$ and existing tools for the analysis of stochastic gradient descent for non-convex functions.

**Theorem 4.4** (Formal Theorem 1.1, part (b)). *Let $\mathcal{G}$ be a MPG and let $s_0 \in \mathcal{S}$ denote an arbitrary initial state. Let $A_{\max} = \max_i |\mathcal{A}_i|$, and set the number of iterations to be $T = \frac{48(1-\gamma)A_{\max}D^4S^2\delta^4}{\epsilon^6\gamma^3}$ and the learning rate (step-size) to be $\eta = \frac{\epsilon^4(1-\gamma)^3\gamma}{48nD^2A_{\max}^2S\delta^2}$. If the agents run projected stochastic policy gradient (PSGA) starting from arbitrarily initialized policies and using $\alpha$-greedy parametrization with $\alpha = \epsilon^2$, then with probability $1 - \delta$ there exists a $t \in \{1, \ldots, T\}$ such that $\pi^{(t)}$ is an $\epsilon$-approximate Nash policy.*

*Remark* 2 (Weighted and ordinal MPGs). We conclude this section with a remark on Weighted and Ordinal MPGs (cf. Definition in 1). It is rather straightforward to see that our results carry over for WMPGs. The only difference in the running time of (PGA) is to account for the weights (which are just multiplicative constants).

By contrast, the extension to OMPGs is not immediate and the reason is that we cannot prove any bound on the smoothness of $\Phi$ in that case. Therefore, we cannot have rates of convergence of policy gradient. Nevertheless, it is quite straightforward that (PGA) converges asymptotically to critical points (in bounded domains) for differentiable functions. Thus, as long as $\Phi$ is differentiable, it is guaranteed that (PGA) will asymptotically converge to a critical point of $\Phi$. By Lemma 4.1, this point will be a Nash policy.

## 5 Experiments: Congestion Games

We next study the performance of policy gradient in a general class of MPGs that are congestion games at every state (cf. [4]). The setting of the current experiment is illustrated in Figure 4.

**Experimental setup.** There are $8$ agents, $4$ facilities and $2$ states: a *safe* state and a *distancing* state. In both states, all agents prefer to be in the same facility with as many other agents as possible *(follow the crowd)* [12]. In particular, the reward of each agent for being at facility $k = A, B, C, D$ is equal to a predefined positive weight $w_k^{\text{safe}}$ times the number of agents at that facility. The weights satisfy $w_A^{\text{safe}} < w_B^{\text{safe}} < w_C^{\text{safe}} < w_D^{\text{safe}}$, i.e., facility $D$ is the most preferable by all agents. If more than $4 = N/2$ agents find themselves in the same facility, then the game transitions to

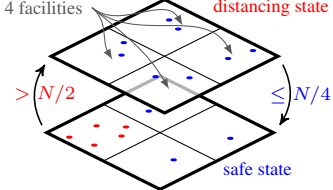

Figure 4: The 2-state MPG.

the distancing state. At that state, the reward structure remains the same, but the weights are reduced by a constant factor, i.e., $w_k^{\text{dist}} = w_k^{\text{safe}} - c$, where $c > 0$ is a (considerably large) constant. To return to the safe state, the agents need to achieve maximum distribution over the facilities, i.e., no more than $2 = N/4$ agents may be in the same facility.

To see that this MDP is a MPG, it suffices to check that every state is a potential game and that condition C2 (i.e., equality of individual dummy terms) of Proposition 3.2 is satisfied. The first claim is straightforward since at each state, the agents play a congestion game [22, 25]. The second claim follows from the fact that the rewards of all agents in all facilities at the distancing state are shifted by the same constant amount, $c$.

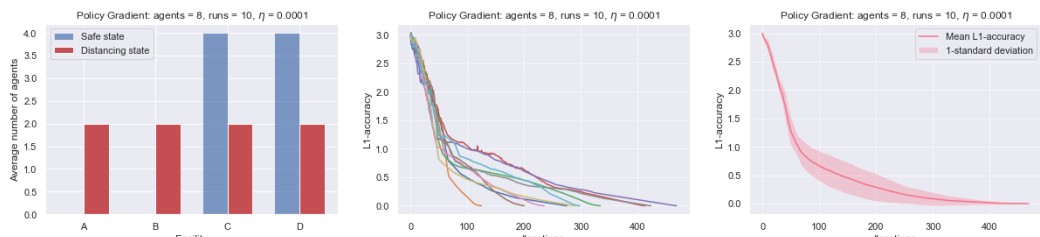

Figure 5: Policy gradient in the 2-state MPG with 8 agents of Section 5. In all runs, the 8 agents learn one of the deterministic Nash policies that leads to the optimal distribution among states (left). Individual trajectories of the L1-accuracy and averages (with 1-standard deviation error bars) show fast convergence in all cases (middle and right columns).

**Paremeters.** We perform episodic updates with $T = 20$ steps. At each iteration, we estimate the policy gradients using the average of mini-batches of size 20. We use $\gamma = 0.99$ and a common learning rate $\eta = 0.0001$ (this $\eta$ is (several orders of magnitude) larger than the theoretical guarantee, $\eta = \frac{(1-\gamma)^3}{2\gamma A_{\max} n} \approx 1e - 08$, of Theorem 4.2). Experiments with randomly generated learning rates (different for each agent), non-deterministic transitions between states and with different weights at each facility in the distancing state (that result in non- MPG structure) produce qualitatively equivalent results and are presented in Appendix E.

**Results.** The left panel of Figure 5 shows that the agents learn the expected Nash profile in both states in all runs. Importantly, this (Nash) policy profile is *deterministic* in line with Theorem 4.2. The panels in the middle and right columns depict the L1-accuracy in the policy space at each iteration which is defined as the average distance between the current policy and the final policy of all 8 agents, i.e., L1-accuracy $= \frac{1}{N} \sum_{i \in \mathcal{N}} |\pi_i - \pi_i^{\text{final}}| = \frac{1}{N} \sum_{i \in \mathcal{N}} \sum_s \sum_a |\pi_i(a \mid s) - \pi_i^{\text{final}}(a \mid s)|$.

# 6 Further Discussion and Conclusions

We presented positive results (both structural and algorithmic) about the performance of independent policy gradient in Markov Potential Games (MPGs). We showed that MPGs always possess deterministic Nash policies and that independent policy gradient is guaranteed to converge (polynomially fast in the approximation error) to (deterministic) Nash policy profiles even in the case of finite samples (assuming a direct parameterization with greedy exploration). Our definition of MPGs generalizes prior works on state-based potential MDPs (importantly, by encompassing MDPs that are not necessarily potential at each state) and demonstrates the effectiveness of simultaneous policy gradient in learning Nash policies even without the need to impose additional assumptions on state-based potential functions (cf. [16, 32]). Given these positive results, several interesting questions emerge.

**Open questions.** When it comes to online learning in normal form potential games, it is possible to prove that many naturally motivated dynamics converge to deterministic Nash equilibria with certain desirable stability properties for most initial conditions [13, 24, 7, 17]. To produce such equilibrium selection results, standard Lyapunov arguments do not suffice and one needs to apply more advanced techniques such as the Center-Stable-Manifold theorem [15]. Studying such techniques in the context of MPGs is a fascinating direction for future work.

On the other hand, given the complexities of multi-agent, state-based environments, it is highly unlikely to expect that practical algorithms can always guarantee convergence to equilibrium. This is already the case even for the more restricted settings of normal-form games [34, 2]. Nevertheless, deriving strong theoretical guarantees in the sense of cyclic/recurrent orbits, invariant functions [18] or social welfare [31] in the context of exact, weighted or ordinal MPGs is another stimulating direction for future work. As a measurement of the inefficiency due to lack of coordination between agents, it would also be interesting to perform a Price of Anarchy type of analysis [14] as has been excessively done in the context of normal-form potential (congestion) games (e.g., [26]).

Finally, other natural directions for future work involve the study of policy gradient or variations thereof (such as Natural Policy Gradient) in MPGs under different policy parametrizations, cf. [1], or the study of settings that fruitfully combine tools from both cooperative and competitive settings (as in [10, 36, 38]) that have (up to now) produced results in orthogonal directions.

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
