# OpenReview forum: "Global Convergence of Multi-Agent Policy Gradient in Markov Potential Games"
_NeurIPS.cc/2021/Conference — NeurIPS 2021 Submitted_

### Official Review · Reviewer_7dwv · 2021-07-16

**Rating:** 5
**Confidence:** 3

**Summary:**

This paper considers a multi-agent coordination setting as all agent utilities are perfectly aligned with each other via a common potential function. The authors claim that they present a novel function of Markov Potential Game (MPG) and prove fast convergence of independent policy gradient to Nash policies.

**Limitations And Societal Impact:**

The authors provide open questions.

**Main Review:**

Potential game is one of the key concepts in multiagent reinforcement learning, especially when the cooperative setting is assumed. The main question of this paper is fast convergence guarantees for multi-agent RL, and the reviewer believes this is an important question. Even though this paper has a well-designed direction, however, the reviewer thinks the current version of the manuscript is not ready to be published at NeurIPS.


1. Are the findings meaningful for MARL researchers?

Theorem 1.2 is an important finding but it needs more discussion whether it is meaningful results in the perspective of multi-agent reinforcement learning. First of all, Theorem 1.2(a) is trivial unless it provides uniqueness. Next, Theorem 1.2(b) is a meaningful finding so we need Theorem 1.2(c) to find conditions for an MDP to be an MPG. The conditions for it are provided in Proposition 3.2. C1 claims that the state transition function should not depend on the action. In this case, RL is not necessary since RL is the effort to find an optimal action.
The reviewer believes that C2 is a meaningful part, leading that we need immediate reward which only includes the potential term. However, is it easy to decouple the reward to common term and dummy term? Most cooperative MARL problems have coupled reward functions. It would be beneficial to provide examples of reward function of standard MARL benchmarks (e.g., Starcraft, Hanabi, OpenAI hide-and-seek, etc).


2. Can the main document stand alone?

The authors simply provide a long version document (31 pages) as supplementary material where a lot of parts are overlapped with the main document. If they separate the contents into main document and supplementary material, the main paper would be easy to follow, and supplementary material will be helpful. Currently, it is hard to find a piece of proper information in supplementary material since it is another long document.
In addition, there are a lot of never-defined notations in the main document. In the current format, the main paper cannot stand alone without supplementary, which is not desirable.



**Time Spent Reviewing:**

3

---

> ### Author Response · Authors · 2021-08-10
> **Response to Reviewer 7dwv's official review**
>
> We thank Reviewer 7dwv for their constructive feedback on our paper.
>
> Question 1.
>
> Concerning the relevance of our results to MARL researchers and in particular, the importance of Theorem 1.2, we appreciate the opportunity to clarify the following. Theorem 1.2(a) is not trivial since MPGs refer to multi-agent interactions for which little is known. In fact, MPGs constitute the first (to the best of our knowledge) class of multi-agent MDPs with provably deterministic (pure) Nash policies. This is in contrast to single-agent MDPs, in which indeed, this result is known. Uniqueness is impossible to establish in such a large class of games since even normal-form potential games (i.e., games without state) may have multiple equilibria: as it is classically known even in the simplest cases of (stateless) potential games, i.e., potential games with two agents and two actions, such as e.g., Battle of the Sexes, multiple Nash equilibria exist. In fact, these multiple Nash equilibria have distinct payoffs for each agent. This is a quintessential property of potential games, which makes them strictly harder than single agent optimization as well as two-agent zero-sum games.
>
> To elaborate further on the relevance of the current research to current MARL researchers, we note that while we provide some sufficient conditions for MDPs to be MPGs, clearly these conditions these are far from necessary. In particular, as our Example 3 showcases, we can have much more complex settings that are MPGs but which are not potential at every state. This shows that MPGs can be constructed from complex building blocks and can, thus, capture complex multi-agent interactions. Such interactions, as exemplified by the settings in the paper "Multi-agent Reinforcement Learning in Sequential Social Dilemmas (by Deepmind)", by Hanabi or by the Capture the Flag game (Capture the Flag: the emergence of complex cooperative agents) just to name a few, are in the frontier of current research. Nevertheless, most existing results are heuristic and providing formal guarantees is challenging. Such settings may not have an exact potential and may be better captured by perturbed versions of exact MPGs, in particular, by ordinal or weighted MPGs. To address this issue, we provide experiments in such settings but more importantly, we show that our theoretical guarantees naturally apply to these settings as well. In short, we hope that our theoretical results complement these heuristic findings and provide a theoretical justification for the derivation of the positive empirical results.
>
> Question 2.
>
> We appreciate the reviewer's comments regarding the readability of our paper. Our intention was to make the paper as easy to read as possible and we thought that giving a concise overview of our methods and results in the main part and including the full version as supplementary material would optimize the reading experience. We regret to hear that this was not the case for the reviewer and their concerns are noted. We agree that the paper needs to be stand-alone and when proofreading the paper in response to this comment, we indeed found that the definition of, e.g., the mismatch coefficient (D) was missing from the main text.
>
> Thus, this comment is well-received and we are committed to act on this. We are confident that without significant changes in the main part of the paper (we think to reduce the technical overview in the Introduction, to include the 1-2 missing definitions that we spotted and to potentially add more intuition and experiments in the gained space) and by including only the missing parts in the appendix (instead of the full version), we can make its content considerably more easy to read by a wider audience.
>
> Again, we thank reviewer 7dwv for their concise comments to improve our paper.

---

### Official Review · Reviewer_4tX4 · 2021-07-16

**Rating:** 7
**Confidence:** 4

**Summary:**

In this paper, the authors present a new definition of Markov Potential Games (MPG). They discuss the relationship between MPG, normal form potential game, and Markov games where every state-game is a potential game.  They study MPGs by presenting sufﬁcient conditions for MPGs and the existence of deterministic Nash policies.  They show the conference of independent-playing policy gradient to Nash policies, in both exact gradient case and finite sample case. Experiment results validate the congergence.

**Ethical Concerns:**

No visible ethical issues.

**Limitations And Societal Impact:**

No visible negative societal impact.

**Main Review:**

- Significance:
MPG serves as an important and fundamental setting towards understanding multi-agent systems. Understanding the behavior of standard methods such as policy gradient is of central interest in the relevant literature. Hence this paper provides a solid contribution.

- Quality and Clarity:
The authors have conducted a clear theoretical analysis, which seems to be very rigorous. In the experiments, the authors conducted congestion games to validate the predicted convergence clearly.

- Originality:
The definition of MPG in the present paper is novel to my knowledge.  The proof of convergence via gradient dominance is commonly used. To improve the originality, I believe the author could discuss more what's the difference between MPG defined in this paper and in some other papers, for example:

> Macua, Sergio Valcarcel, Javier Zazo, and Santiago Zazo. "Learning parametric closed-loop policies for markov potential games." arXiv preprint arXiv:1802.00899 (2018).

Overall, this paper is worthy of acceptance to me. I'm glad to increase my score if the discussions are provided and more experiments on some other examples of MPGs could be conducted.

- Minor points: In equation (3), should the $\alpha_i$ be $\alpha$? A similar typo exists in the full version too (eq (4)).

**Time Spent Reviewing:**

3

---

> ### Author Response · Authors · 2021-08-10
> **Response to Reviewer 4tX4's official review**
>
> We thank Reviewer 4tX4 for their positive evaluation of our paper and their useful comments.
>
> Concerning 4tX4's suggestion, we appreciate the opportunity to discuss the differences between our definition of MPGs and the one in the suggested paper (henceforth referred to as [32] following its number in the references of our submission). MPGs in [32] are defined in equation (15). This equation involves a difference of terms within the expectation operator (in response to a change in an agent's policy) rather than a difference between expectations. While seemingly minor, this difference is fundamental: the expectation here is over trajectories, i.e., paths that are induced by agent's actions. Taking the same expectation for two different policies (as in equation (15) of [32]) implies that the change in the policy of the agent does not affect the expected trajectory. But this is precisely the crux of state-based interactions: in MDP settings (unlike normal-form games), actions affect transitions and a minor change in an agent's policy may have far-reaching effects in their long-term value.
>
> To clarify, action-independent transitions constitute a sufficient condition (as we show in C1 of Proposition 3.2) to obtain an MPG. However, they represent a considerably more restrictive condition than the definition of MPGs in the current paper.
>
> To further elaborate on that, the proposed definition of [32] seems to cover a subset of state-based potential games that, in addition, have action independent transitions. As Theorem [2] in [32] suggests such MDPs involve games that are potential at every state (equation (25)) and for which the dummy term does not depend in any way in the policy of the agent (equation (26)). As we show in Example 2, this is precisely the problem: when transitions depend on agent's actions (as in many non-trivial RL settings), the dummy term (of the state-based normal form potential game) becomes dependent to that actions (thus, it is not a dummy term anymore). In turn, as our Example 3 shows, the current definition includes MDPs that are not potential at every state.
>
> In defense of [32]'s definition (which coincides with case C1 in our Proposition 3.2), agent independent transitions may actually occur in RL settings in a very interesting (non-trivial) case. In particular, as such systems grow larger, transitions may depend on aggregate network quantities and not on individual actions. If each agent has only a negligible influence on aggregate outcomes (e.g., self-interested actors in large economies, independent particles in organisms or individual agents in large AI systems), then condition C1 in Proposition 3.2 (equivalently, definition of MPGs in [32]) can be seen to apply in the limit (i.e., approximately, but with high accuracy as the number of agents grows larger). Thus, these properties can be useful in the experimental study of such (large-scale) systems by providing theoretical convergence guarantees for policy gradient methods (current paper) or by providing analytical ways to determine their equilibria via optimization ([32]).
>
> Related to the previous comment, the reviewer's suggestion to conduct additional experiments gave us a very constructive idea for a useful addition to the current paper: [32] study the "Great Fish War" game which is an MPG according to [32]'s definition (thus, also an MPG according to our definition and Proposition 3.2, C1). This game is analytically studied in [32] and thus, it provides a perfect test-bed to run our algorithmic methods (policy gradient). We did this in response to the reviewers suggestion (for which we are thankful), and we can report fast convergence to the (stationary) equilibrium policy (weight). This result can be included in our paper since it demonstrates a very efficient (algorithmic) method to determine the equilibria of the game which complements the analytic approach of [32].
>
> In addition to the above, we can also report in a more extended appendix,  experimental results in ordinal MPGs that comprise multiple states with Battle of the Sexes (a potential 2x2 game) played at every state. In our current submission, we included the state-based congestion game instead as an example that can include more agents.
>
> Finally, concerning the minor point, we thank the reviewer for catching this (it should be $\alpha$ instead of $\alpha_i$, correct) and we will fix this in the updated version.

---

### Official Review · Reviewer_GJfn · 2021-07-16

**Rating:** 5
**Confidence:** 4

**Summary:**

The paper focuses on one of the game theoretic views of Reinforcement Learning: the convergence of policy gradient in games of several agents. The authors build on the existing theory by adding new definitions that support the multi-agent learning systems in order to measure the complexity of the convergence of the policy gradient to Nash policies.

**Limitations And Societal Impact:**

The paper builds on [10] of the included literature, where the authors had introduced the first independent policy gradient algorithms for competitive reinforcement learning in zero-sum stochastic game. The main result of the current paper is the fast convergence of these algorithms to the so-called 'Nash policy'.

1) This seems an interesting result but It seems to make a minor contribution to a reasonable problem. It looks like a simply developed extension of the previous paper. Its novelty and significance as far as the paper's modelling is concerned (as a whole) needs to be validated.
From a first point of view, the results do not seem like a breakthrough.

2) The 'novel definition of MPGs that generalizes prior attempts' is interesting from the theory point of view, but it seems that it is specially developed here so that it creates a desired context. That is, it seems like a theoretical construction which supports the result, rather than solving a core problem that had arisen in [10].

3) Also, although is it a well suited result for the theory of Markov Decision Processes (MDPs) founded in [10], I am not certain of how significant it is for the readers of this conference. The focus is mainly game-theoretic and about the complexity of this game, rather than in the theory of Learning. The theorems concern the relation of Markov Potential Games to (MDPs) and conclude the convergence to an \epsilon-Nash policy. This sounds like a Game Theory result.

Which are the learning systems that benefit from the fast convergence?
Are there any 'common' machine learning systems that benefit from this?

4) The authors list an example of an eight agent congestion game which is demonstrating the MPG property. What is the applicability of this example and their convergence result in a learning context with several agents? Have the authors tried to apply that somewhere in machine learning? Can they present this in the paper?



**Main Review:**

The paper is written in a clear and formal manner. The conclusions seem to respect the stated theorems (as corollaries). The next section has some points about the specificity of the paper's context and its novelty results.

**Time Spent Reviewing:**

25

---

> ### Author Response · Authors · 2021-08-10
> **Response to Reviewer GJfn's official review**
>
> We thank Reviewer GJfn for their constructive feedback on our paper. As many of the comments in this review are with respect to reference [10], which analyzed and proved convergence for independent policy gradient in two-player Markov zero sum games, we appreciate the opportunity to highlight the key differences between that paper and ours.
>
> Question 1.
>
> Concerning modelling, there are two fundamental differences between our paper and [10]: we consider *multi-agent Markov potential games* i.e., multi-agent interactions with aligned incentives (coordination) whereas [10] consider *two-agent Markov zero-sum games*, i.e., two-agent interactions with directly opposing incentives (competition). Thus, the first main difference is that our paper considers settings for $n$ agents rather than just 2 as in [10] or single-agent as in [1]. Accordingly, our paper makes a significant leap forward in the study of multi-agent settings that was missing from the literature. The second main difference is that such settings (i.e., cooperation vs competition) can be radically different in terms of learning dynamics (among others) as recent theoretical and experiment research has clearly demonstrated (we explain this in the Introduction).
>
> Concerning the similarities/differences in the technical parts, we clarify the following. Both papers use policy gradient and thus crucially rely on the gradient dominance properties of [1] (Agarwal et al.). Yet, our analysis goes beyond that of [10] among others by (1) utilizing key aspects of the potential function to overcome the issue of equilibria with different values (a problem that is not present in two-agent, zero-sum games), by (2) proving the existence of an agent-wise Gradient Domination Property and by (3) allowing agents to update their policies independently and simultaneously using the same learning rate (whereas as the authors of [10] mention in their paper, their results require the two-agents to agree to use different learning rates (equivalently, to take turns), which hampers the notion of "independent").
>
> Finally, concerning our results, our contribution with respect to the existing literature (including mainly [1], [10] and [33]) is that we obtain the first provable guarantees for the convergence of a standard RL algorithm (policy gradient) in a general multi-agent setting (allowing for arbitrary $n\geq 2$ agents) that goes beyond single agent settings (se [1]), team games (in which all agents receive the same rewards) (see [33]), or games that have the “min-max equals max-min” property that is crucial to the results for the two-player zero sum setting (see [10]).
>
> Question 2.
>
> Concerning the definition of MPGs, we are firm that there is nothing special about this definition to make the math work. In fact, the contrary holds. We naturally extended the normal-form definition of potential games to the state-based setting. In doing so, we did not add any artificial/special assumptions on the transitions or utility (value) functions of the agents as some previous attempts did (which we, thus, generalize).
>
> Concerning the motivation to study this setting, we contend that the MPG setting was a natural direction calling to be addressed rather than a choice based on applicability of the methods, and in particular it does in fact address a core problem raised in [10] (and in [1] which we consider to be the second base-paper along with [10] for our study). In the *Future directions* section of [10] the authors write: *Many games of interest are not zero-sum, and may involve more than two players or be cooperative in nature. It would be useful to extend our results to these settings, albeit likely for weaker solution concepts.*
> Our paper addresses exactly this direction; we go beyond zero-sum, beyond two players, and work towards an understanding multi-agent MDPs that are cooperative in nature. And in fact we are able to do this without using a weaker solution concept.
>
> Question 3.
>
> Concerning the interest of this research to the audience of NeurIPS, we are firm that the paper is relevant to NeurIPS and that its content appeals to both theorists and applied researchers working in the area. First, we would like to clarify that the result concerns the complexity (rate of convergence) of the learning method (policy gradient) rather than the complexity of the game itself. Second, the Nash equilibrium is the standard solution concept of MDPs with multiple agents (see references in the paper and below). Finally (and most importantly), MDPs and Game Theory (in particular, the theoretical foundations of learning dynamics in multi-agent interactions) are currently studied by many RL groups that are consistently publishing in NeurIPS (or similarly themed conferences). An indicative (and highly non-exhaustive) list is provided below:
>
>     https://proceedings.neurips.cc/paper/2017/file/3323fe11e9595c09af38fe67567a9394-Paper.pdf
>     https://papers.nips.cc/paper/2002/file/f8e59f4b2fe7c5705bf878bbd494ccdf-Paper.pdf
>     https://deepmind.com/blog/article/capture-the-flag-science
>     https://deepmind.com/blog/article/EigenGame
>     https://link.springer.com/chapter/10.1007/978-3-642-27645-3_14
>     https://arxiv.org/pdf/2011.00583.pdf
>
> Question 4.
>
> We thank Reviewer GJfn for raising this point since it gives us the opportunity to highlight the numerous connections between our experimental and theoretical settings and ML. Recent research reveals a constantly growing network of connections between game theory (congestion games in particular) and ML. Results in this line comprise Deepmind's awarded paper "EigenGame: PCA as a Nash Equilibrium", "Learning Pure Nash Equilibrium in Smart Charging Games", "Online Learning of Nash Equilibria in Congestion Games" and "Deep Neural Networks Are Congestion Games: From Loss Landscape to Wardrop Equilibrium and Beyond", just to name a few.
>
> Our experiment also applies to arbitrary numbers of agents (we included an instance with 16 agents in the supplementary material to demonstrate this, but clearly larger simulations are possible) and convergence is still guaranteed to be polynomially fast in the approximation error (by our main Theorem) for any number of agents. As showcased by the above references, congestion games are ubiquitous in ML research and their applications are only growing.
>
> We hope that this clarifies. In any case, the comment is well-received and we will be happy to add such links between our work and ML research in our updated version.

---

### Official Review · Reviewer_HF6B · 2021-07-16

**Rating:** 7
**Confidence:** 3

**Summary:**

The paper introduces a novel class of Markov Games that generalize classical potential games in the normal-form representation. They show that such games always admit deterministic Nash policies, and that they enjoy some interesting properties that they make them a more suitable generalization of potential games than other attempts in the literature. The main result of the paper is that, in Markov Potential Games, policy gradient algorithms converge to Nash. The authors also provide an experimental analysis on congestion games.

**Limitations And Societal Impact:**

Adequate.

**Main Review:**

Originality: I think that the problem studied by authors is interesting, and that the paper makes a significant contribution to the research area. The idea behind the definition of Markov Potential Game is novel, and I think it provides a nice and clean generalization of its normal-form counterpart.

Quality: As far as I am concerned, the technical claims are correct.

Clarity: The paper is well written, but I think that some modifications would help clarify it. For instance, the authors could better clarify the difference between (a) and (b) in Theorem 1.1 at the beginning, since I needed to step forward to the formal result later in the paper to better figure out what was going on. Also, the technical review is quite dense and hard to follow. Maybe, it's better to do not go into technicalities in the Introduction and add some more intuitions when the technical part comes into play.

Significance: I think that the paper would have a good fit in the NeurIPS conference.

**Time Spent Reviewing:**

2

---

> ### Author Response · Authors · 2021-08-10
> **Response to Reviewer HF6B's official review**
>
> We thank Reviewer HF6B for their positive evaluation of our paper and their useful comments.
>
> We agree that the phrasing of Theorem 1.1 can be made more clear in order to highlight the key difference between (a) and (b). In particular, (a) refers to the exact gradients case and (b) refers to the finite sample gradients case, respectively. It is a good idea to add this clarification in Theorem 1.1 and also to add pointers to Theorems 4.2 (case (a)) and 4.4 (case (b)) directly after Theorem 1.1.
>
> We also agree that the technical review is quite dense, while also being quite lengthy. We thank the reviewer for their suggestion (which we will follow) to move some parts directly into the technical sections or to remove them completely and use the added space to expand more in other areas (e.g., experiments) of the paper.

---

### Author Response · Authors · 2021-08-26
**Feedback**

We would like to thank you for your hard work. We just wanted to reach out and see if any of the reviewers had any comments back to our rebuttal. We are looking for feedback on whether the points made in the reviews have now been addressed. We are happy to answer any remaining questions.

---

### Decision · Program_Chairs · 2021-09-27

**Decision:**

Reject

**Comment:**

The discussion of this paper raised a lot of points that, I hope, will be useful to the authors:
1. there is no concern on the mathematical derivation; this is a very good selling point, but it is not sufficient
2. while there is a strong agreement that this paper would perfectly fit with more game-theory oriented venues, there are objections that, in the current form, it fits with ML venues: in particular, the authors should strengthen the learning theory part. As remarked by some Reviewers and argued in the rebuttals by the authors, some parts of the paper should be re-organized, thus requiring a second round of reviewing.

My suggestion to the authors is to implement the comments raised by the Reviewers carefully. The converge result is nice and a bit of effort spent to re-organize the paper will lead to a very probable acceptance in a top-rated ML conference.